# Successful Treatment of a Patient with Cardiac Arrest Due to Hyperkalemia by Prolonged Cardiopulmonary Resuscitation along with Hemodialysis: A Case Report and Review of the Literature

**DOI:** 10.3390/medicina57080810

**Published:** 2021-08-07

**Authors:** Nuri Kose, Ferruh Bilgin

**Affiliations:** 1Department of Cardiology, Yucelen Hospital, Mugla 48000, Turkey; 2Department of Anesthesiology and Reanimation, Yucelen Hospital, Mugla 48000, Turkey; ferruhbilgin@yahoo.com

**Keywords:** hyperkalemia, kidney failure, cardiac arrest, hemodialysis

## Abstract

Severe hyperkalemia is a potentially life threatening cardiac emergency, especially in patients with renal failure, and can lead to fatal arrhythmias such as ventricular fibrillation or asystole, leading to cardiac arrest. We report a case of a 39-year-old woman who developed sudden cardiac arrest secondary to hyperkalemia (9.95 mEq/L) with renal insufficiency. Despite 20 min of cardiopulmonary resuscitation (CPR) and conventional treatment for hyperkalemia, the cardiac arrest persisted. Hemodialysis was then initiated via the right femoral vein during CPR, and the patient restored spontaneous heartbeat 40 min later. Hemodialysis should be considered in the course of CPR in severe hyperkalemia induced cardiac arrest if conventional therapies fail.

## 1. Introduction

Hyperkalemia is one of the few potentially lethal electrolyte disorders. Acute hyperkalemia is commonly defined as a serum potassium level of >6.0 mEq/L. Hyperkalemia occurs most frequently in patients with chronic kidney disease (CKD). Other important risk factors for hyperkalemia are the use of drugs that interfere with the renin angiotensin aldosterone system, hypertension, congestive heart failure, coronary artery disease, diabetes mellitus, liver disease, and adrenal insufficiency, as well as acute conditions such as metabolic acidosis, recent blood transfusions, and tissue necrosis [1].

Hyperkalemia reduces the resting membrane potential in both striated and smooth muscle, which leads to increased cardiac depolarization and muscle excitability. The resulting electrocardiographic (ECG) changes do not follow a consistent, stepwise pattern; however, and they may occur rapidly. The risk of arrhythmias increases with potassium values >6.5 mEq/L. Small elevations above this value can cause peaked T waves to quickly progress to ventricular fibrillation or asystole. The longer a patient has elevated potassium concentrations, the greater the risk of sudden deterioration [2]. Therefore, the quality and quantity of food intake is very important in patients with renal failure.

## 2. Case Report

A 39-year-old female patient was admitted to an emergency unit with nausea, debility, and weakness in legs. She had been undergoing treatment of hemodialysis due to end-stage chronic kidney disease for three days per week for the past4 years. She had performed her last dialysis via arteriovenous fistula on the left forearm 48 h before admission. No family history of kidney failure was found in the anamnesis. ECG monitoring was initiated. Arterial blood pressure was 100/70 mmHg from the right arm, and the heart rate was 100 bpm from the right radial artery. The ECG curve revealed a heart rate of 102 bpm, a QRS time of 140 milliseconds, a flattened p wave, and a spiked T wave (Figure 1a). These ECG findings suggested hyperkalemia. Laboratory analyses performed in the emergency department included a complete blood cell count, metabolic panel, kidney panel with potassium, and blood–gas analysis. Blood–gas analyses from the right radial artery revealed that the pH was 7.29, pCO_2_ was 33 mmHg, and pO_2_ was 88 mmHg. Her serum potassium concentration was 9.95 mEq/L (3.60–5.00 mEq/L), blood urea nitrogen levels were 216 mg/dL (10–50 mg/dl), creatinine was 8.69 mg/dL (0.4–1.4 mg/dL), sodium level was 137 mmol/L (135–147 mmol/L, calcium level was 6.3 mg/dL (8.6–10.8), chloride was 103 mmol/L (98–111 mmol/L), white blood cell count was 7.9 × 10^3^ cells/µL (4.0–11.0 × 10^3^ cells/µL) with 84% segmented neutrophils (50−80%), and hemoglobin level was 11.7 g/dL (usually 11–18.8 g/dL for women). 

Shortly after admittance, ventricular fibrillation and cardiac arrest developed. Standard cardiopulmonary resuscitation (CPR) was commenced. Simultaneously, 30 mL of calcium gluconate (10%) was administered for 5 min, 50 mEq of sodium bicarbonate was administered for 5 min, and glucose and insulin (50 mL of 50% dextrose and 10 U regular insulin) were administered intravenously for 15 min. Despite 20 min of cardiopulmonary resuscitation and conventional treatment for hyperkalemia, the cardiac arrest persisted, and normal cardiac rhythm was not recovered. Veno-venous hemodialysis was initiated via the right femoral vein due to hyperkalemia. CPR was continued and the blood flow (150–200 mL/min) was adequate. Heparin (5000 IU) was administered intravenously. Hemodialysis was performed in 4 h with a dialysate fluid that contained 1% potassium. Spontaneous sinus rhythm and blood pressure were restored at the 40th minute of hemodialysis and at the 60th minute of total CPR. After the return of spontaneous circulation, serum potassium levels were 5.2 mmol/L, systemic arterial blood pressure. 110/70 mmHg, ECG exhibited a sinus rhythm, the peak of the T wave decreased, and the pulse was 98 bpm (Figure 1b). After 30 min, the subject regained her consciousness and was extubated. Subsequently, 4 h of hemodialysis was performed for the following consecutive 3 days.

The laboratory analyses from the previous hemodialysis sessions revealed ineffective hemodialysis. The insufficiency of the arteriovenous fistula was the underlying cause of this situation. The subject had three arteriovenous fistulas previously due to insufficiencies. A right subclavian vein hemodialysis catheter was inserted before discharge. An effective regimen of 12 h per week of hemodialysis was planned and the potassium levels were monitored frequently thereafter. The subject had no subsequent problems during follow-ups, however she died due to fatal pneumonia after 5 years. 

## 3. Discussion

Potassium is a ubiquitous cation contained mostly within the intracellular fluid; only about 2% of the total body potassium is found in the extracellular fluid. In healthy humans, serum potassium levels are tightly controlled within the narrow range of 3.5 to 5.0 mEq/L, thus retaining a normal ratio between the intracellular and extracellular compartments. This homeostasis plays a critical role in maintaining the resting cellular membrane potential and neuromuscular function, and is essential for the normal activity of the muscles, nerves, and heart [3]. Under normal conditions, membranes and potential differences, particularly between cardiac membranes, do not affect the changes in potassium levels. Nevertheless, rapid and significant changes in the serum potassium concentrations may result in potassium shifts and life-threatening outcomes.

Hyperkalemia, resulting from a disproportion of potassium homeostasis, is defined as when serum potassium levels are greater than 5.0 mEq/L, and is further classified as mild, moderate, or severe cases [4]. Severe hyperkalemia (potassium levels of at least 6.5 mEq/L) is a potentially life-threatening electrolyte disorder that has been reported to occur in 1% to 10% of all hospitalized patients, a higher percentage than that seen in outpatients [3,5,6]. A retrospective study of 29,063 patients revealed that hyperkalemia was directly associated with sudden cardiac arrest in seven cases [7].

Severe hyperkalemia may lead to paralysis, paresthesia, decreased deep tendon reflexes, or dyspnea. However, the first sign of hyperkalemia may be a peaked T wave in the ECG [8]. As the serum potassium increases, gradual changes may be seen in the ECG through a reduction in the amplitudes or complete loss of P waves, prolonged PR intervals, shortened QT intervals, and wide QRS complexes. If not treated, sinus wave patterns, idioventricular rhythms, ventricular fibrillation, and asystolic cardiac arrest may develop in the advanced stages of hyperkalemia [9].

Treatment of severe hyperkalemia aims to antagonize the effects of potassium on excitable membranes, forcing potassium to rapidly shift into the cells from being in circulation, and preventing the heart from the effects of hyperkalemia by removing the potassium from the organism. Conventional therapies, including administering intravenous sodium bicarbonate, calcium gluconate, insulin with glucose, and several beta-2 agonists, are commonly employed as temporary measures. These treatment modalities that relocate the potassium and stabilize the cellular membranes are rapid-acting, but transient options. Moreover, medical treatment generally remains inadequate in patients with hyperkalemia, and urgent hemodialysis is often needed [6]. If the cardiac rhythm cannot be recovered with CPR and medical treatment in cases of cardiac arrest due to hyperkalemia, hemodialysis accompanying CPR may be beneficial. During cardiopulmonary resuscitation, external cardiac compression can sustain adequate blood flow for hemodialysis [10,11,12].

There are only a limited number of case reports concerning patients who have suffered cardiac arrest due to hyperkalemia and have not responded to the CPR and medical treatment, but recovered after the enforcement of CPR and hemodialysis. In one such case, Gütlich et al. [10] had restored spontaneous circulation in the first 30 min through to administration of CPR and hemodiafiltration in a patient who had suffered hyperkalemic (11.4 mEq/L) cardiac arrest. In another case report by Kao et al. [11], a 68 years-old female patient who suffered cardiac arrest due to renal failure and hyperkalemia did not respond to 100 min of CPR and conventional treatment, although she had recovered a sinus rhythm 20 min after the enforcement of hemodialysis accompanying CPR. Additionally, Lin et al. [12] reported that they recovered the sinus rhythm in three cases of hyperkalemic cardiac arrest with the administration of hemodialysis during CPR. Successful treatments in these cases are rarely reported in the literature; however, the clinical characteristics and treatments are presented here in Table 1.

## 4. Conclusions

Hyperkalemia is a common occurrence, especially in patients with CKD, and it is associated with increased mortality. Hemodialysis should be considered as a rescue method for hyperkalemic cardiac arrest if standard CPR and medical treatments of hyperkaliemia are not effective.

## Figures and Tables

**Figure 1 medicina-57-00810-f001:**
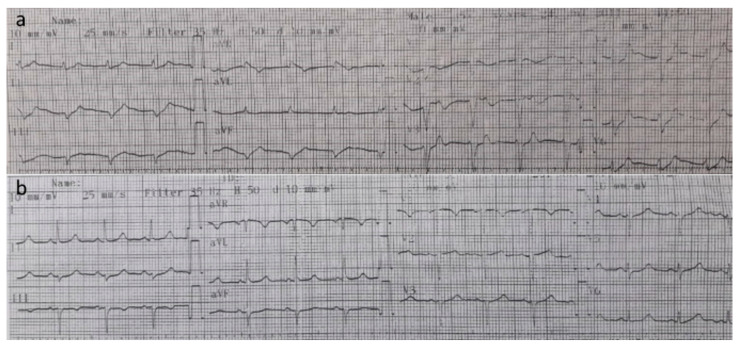
(**a**) ECG at admission. (**b**) ECG after 4 h of hemodialysis. (ECG: Electrocardiography).

**Table 1 medicina-57-00810-t001:** Clinical outcomes of patients with hyperkalemic cardiac arrest with hemodialysis during CPR in the literature.

Study	Age (Years)	Sex	Arrest Rhythm	Serum K^+^ Level at Arrest (mmol/L)	Initial Treatment	CPR Pre-Hemodialysis (min)	CPR Time along with Hemodialysis (min)	Serum K^+^ Level at Return of Spontaneous Circulation (mmol/L)	In-Hospitalmortality
**Gomez-Arnau et al.** [13]	36	M	Asystole	9.7	Epinephrine, calcium chloride, sodium bicarbonate, insulin with glucose, propranolol, and lidocaine	70	75	6.6	No
**Torrecilla et al.** [14]	53	M	Asystole	10.2	Diuretics, calcium chloride, sodium bicarbonate, and insulin with glucose	N/A	90	6.5	No
**Lin et al.** [12]	27	M	VF	9.6	Epinephrine, sodium bicarbonate, calcium chloride, insulin with glucose, and dopamine	55	25	7.6	No
	58	M	VF	8.5	Epinephrine, sodium bicarbonate, calcium chloride, and insulin with glucose	35	30	7.2	No
	77	F	Asystole	8.5–10.5	Epinephrine, sodium bicarbonate, calcium chloride, insulin with glucose, xylocaine, and procainamide	105	35	5.2	Yes (died a few hours later)
**Costa et al.** [15]	57	M	Asystole	9.6	Epinephrine, sodium bicarbonate, calcium chloride, and insulin with glucose	40	95	7.2	Yes (died (after 3 day)
**Kao et al.** [11]	68	F	Asystole	8.3	Epinephrine, sodium bicarbonate, calcium chloride and insulin with glucose	100	20	5.1	No
**Gütlich et al.** [10]	64	F	Asystole	11.4	Adrenalin, noradrenalin, atropine, sodium bicarbonate, calcium chloride, and insulin with glucose	N/A	30	7.0	No
**Iwanczuk et al.** [16]	53	M	N/A	8.5	Epinephrine, sodium bicarbonate, calcium chloride, and insulin with glucose	N/A	40	5.4	No
**Kose and Bilgin (presentcase)**	39	F	VF	9.95	Epinephrine, sodium bicarbonate, calcium chloride, and insulin with glucose	20	40	5.2 (after hemodialysis)	No

CPR, cardiopulmonary resuscitation; F, female; M, male; K^+^, potassium ion; min, minute; N/A, not available; VF, ventricular fibrillation.

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
