# Peer review of "Successful Treatment of a Patient with Cardiac Arrest Due to Hyperkalemia by Prolonged Cardiopulmonary Resuscitation along with Hemodialysis: A Case Report and Review of the Literature"

_medicina, 2021, doi:10.3390/medicina57080810_

Round 1

Reviewer 1 Report

The case study by Kose and Bilgin entitled "Successful treatment of a patient with cardiac arrest due to hyperkalemia by prolonged cardiopulmonary resuscitation along with hemodialysis: a case report and review of the literature" presents a clinical case of a 39-year-old woman who developed sudden cardiac 15 arrest secondary to hyperkalemia with renal insufficiency. The authors proposed hemodialysis in the course of CPR in severe hyperkalemia induced cardiac arrest if conventional therapies fail.

The case report is well presented. However, it is be advisable to provide  family/genetic history of CKD running in the patient's family. Additionally, examining the quality and quantity of food intake by appropriate methods is critical in the management of CKD patients. The case report should also include this information. 

Author Response

Response 1: According to the recommendation of reviewer 1, information was given to the article about whether there was a family history of kidney failure and the importance quality of food intake in kidney failure was emphasized.

Reviewer 2 Report

The manuscript would, in my opinion, require careful proofreading by an english speaking person with focus on interpunction and vocabulary.

line 39 the sentence requires a verb "was admitted?"

line 40 and following: I would consider substitution of: "take dialysis" with another expression like: "was treated, was undergoing treatment..."

line 42: monitoring rather than monitorization

line 44: ECG curve rather than monitorization

lines 49-55: It should be clearly notified that numbers in parentheses are the reference values. Otherwise authors may consider presenting those as a short table.

line 56: adjective "cardiac" should be added to arrest

line 65: "1% of potassium"? Did the authors mean: hemodialysis fluid containing 1 mmol/L potassium?

line 66: the use of ordinal numbers seems inappropriate

line 68: there should be a comment to the T waves in repeated ECG curve

line 69-70: 4 hours of HD daily or 4 in 3 days, this is not clear

Figure 2: is not adding much to the information in article text.

line 90-91: the sentence includes repetition and is not clear.

line: 94: I would consider replacing "imbalance" with another term

line: 101: "flask paralysis" - did the authors mean flaccid paralysis? However the latter is common in polio and botulism cases rather than hyperkalemia.line 109: "rapidly from the circulation" requires a verb

line 123 and 128: "administration" may be replaced by another term.

line 137-138: "a rescue method for hyperkalemic cardiac arrest..."

In addition the use of definite and indefinite articles (a and the) should be reviewed.

Author Response

Response to Reviewer 2 Comments

Point 1: The manuscript would, in my opinion, require careful proofreading by an english speaking person with focus on interpunction and vocabulary.

Response 1: English editing of this case report was made at the request of the reviewer.

Point 2: line 39 the sentence requires a verb "was admitted?".

Response 2: Added ‘’was admitted’’ as a verb to line 39.

Point 3: line 40 and following: I would consider substitution of: "take dialysis" with another expression like: "was treated, was undergoing treatment...".

Response 3: ‘‘Take dialysis’’ was changed ‘‘was undergoing treatment of hemodialysis’’ on line 40.

Point 4: line 42: monitoring rather than monitorization.

Response 4: ‘‘Monitorization’’ was changed ‘‘monitoring’’ on line 42.

Point 5: line 44: ECG curve rather than monitorization

Response 5: ECG monitorization was changed ‘‘ECG curve’’ on line44.

Point 6: lines 49-55: It should be clearly notified that numbers in parentheses are the reference values. Otherwise authors may consider presenting those as a short table.

Response 6: All references are shown in parentheses.

Point 7: line 56: adjective "cardiac" should be added to arrest

Response 7: Before the arrest we added ‘‘cardiac’’ on line 56.

Point 8: line 65: "1% of potassium"? Did the authors mean: hemodialysis fluid containing 1 mmol/L potassium?

Response 8: 1% of potassium is explained ‘‘Hemodialysis was performed in 4 hours with dialysate fluid which contains 1% potassium’’ on line 65.

Point 9: line 66: the use of ordinal numbers seems inappropriate

Response 9: the use of ordinal numbers was corrected.

Point 10: line 68: there should be a comment to the T waves in repeated ECG curve

Response 10: Information about the T wave is given on line 68.

Point 11: line 69-70: 4 hours of HD daily or 4 in 3 days, this is not clear

Response 11: After the first hemodialysis the subject was treated consequtive the first 3 days. ( line 70)

Point 12: Figure 2: is not adding much to the information in article text.

Response 12: Figure 2 was removed.

Point 13: line 90-91: the sentence includes repetition and is not clear.

Response 13: Sentence structure was corrected. (line 90-91)

Point 14: line: 94: I would consider replacing "imbalance" with another term

Response 14: ‘‘imbalance’’ was changed ‘‘disproportion of’’ on line 94.

Point 15: line: 101: "flask paralysis" - did the authors mean flaccid paralysis? However the latter is common in polio and botulism cases rather than hyperkalemia.

Response 15: Flask paralysis was corrected ‘‘paralysis’’. (line 101)

Point 16: line 109: "rapidly from the circulation" requires a verb.

Response 16: forcing potassium to rapidly shift into the cells from being in circulation (line 109).

Point 17: line 123 and 128: "administration" may be replaced by another term.

Response 17: Administration was corrected ‘‘enforcement’’ (line 123 and 128).

Point 18: line 137-138: "a rescue method for hyperkalemic cardiac arrest..."

Response 18: "a rescue method hyperkalemic cardiac arrest..." was changed "a rescue method for hyperkalemic cardiac arrest..." on line 137-138.

Point 19: In addition the use of definite and indefinite articles (a and the) should be reviewed.

Response 19: : The use of definite and indefinite articles (a and the) was corrected. English editing of this case report was made at the request of the reviewer.

Sincerely
